# Quantum Image Steganography Schemes for Data Hiding: A Survey

**Nasro Min-Allah** [1]**, Naya Nagy** [2]**, Malak Aljabri** [3] **, Mariam Alkharraa** [1,*] **, Mashael Alqahtani** [1] **, Dana Alghamdi** [1]**, Razan Sabri** [1] **and Rana Alshaikh** [1]

1   SAUDI ARAMCO Cybersecurity Chair, Department of Computer Science, College of Computer Science and Information Technology, Imam Abdulrahman Bin Faisal University, P.O. Box 1982, Dammam 31441, Saudi Arabia
2   SAUDI ARAMCO Cybersecurity Chair, Department of Networks and Communications, College of Computer Science and Information Technology, Imam Abdulrahman Bin Faisal University, P.O. Box 1982, Dammam 31441, Saudi Arabia
3   Department of Computer Science, College of Computer and Information Systems, Umm Al-Qura University, P.O. Box 715, Makkah 21955, Saudi Arabia
*   Correspondence: 2190005668@iau.edu.sa; Tel.: +966-5068-303-75

**Abstract:** Quantum steganography plays a critical role in embedding confidential data into carrier messages using quantum computing schemes. The quantum variant of steganography outperforms its classical counterpart from security, embedding efficiency and capacity, imperceptibility, and time-complexity perspectives. Considerable work has been carried out in the literature focusing on quantum steganography. However, a holistic view of available schemes is missing. This paper provides an overview of latest advances in the field of quantum-steganography and image-steganography schemes. Moreover, the paper includes discussion of improvements made in the aforementioned fields, a brief explanation of the methodologies used for each presented algorithm, and a comparative study of existing schemes.

**Keywords:** quantum image steganography; high-performance computing; quantum computing; quantum image representations; quantum data hiding





## 1. Introduction

Steganography defines the concept of embedding secret data into a cover message and sending it, such that only the sender and the receiver can learn the content of the embedded message. Thus, any intruder who tries to eavesdrop on the interaction will access the cover data, while the hidden message is protected from being accessed [1]. Quantum steganography, which is a combination of classical steganography [2] and quantum computing, applies the same concept of hiding data using quantum technologies and protocols. A narrower concept, quantum image steganography, involves hiding messages in images and uses quantum image representations as a cover to hide confidential data [3]. In quantum image steganography, the algorithm's success strongly depends on whether the human eye can detect the embedded segment from the cover image. More areas of evaluation need to be considered when it comes to measuring the performance of a quantum steganography scheme.

Different methods of embedding quantum image representations have been adopted by researchers in recent years. One of the most common approaches is the LSBq, short for the least significant qubit approach, which is inspired by the LSB approach (least significant bit) commonly implemented in classical image steganography. The LSB method explains the process of hiding confidential data by embedding each of its bits into the least significant bits of the carrier image; LSBq, on the other hand, adopts the same concept with the focus on embedding qubits rather than bits in quantum image representations [4–6].

One of the most recent implementations for LSBq is by Gabriel et al. [6] who focused on creating a steganography scheme using only the LSBq method to hide a grayscale image in a carrier image represented by NEQR (novel enhanced quantum image representation) [7]. Sahin et al. [5] also used the LSBq approach to implement quantum image steganography in an image modeled by quantum multi-wavelength image representation (QRMW) [8], allowing to hide larger scales of the secret message into the carrier image by varying its wavelengths. Furthermore, the inverted pattern approach applied the LSB method through Luo et al. [9]'s approach to enhance the embedding process in a straightforward and simple execution. Another approach that aimed for a high embedding efficiency is the use of the (CAQW) (controlled alternate quantum walks) protocol for enhancing the procedure of embedding qubits into a cover image, which has shown to produce outperforming results in all areas, including resistance to data loss threats [10].

One of the challenges that researchers face when creating steganographic images is low resolution. Therefore, researchers have incorporated protocols and embedding schemes that preserve the image's quality. For example, the pixel value differencing approach achieved the goal of improving the quality of the embedding procedure as well as preserving the high-resolution state of the cover image by using the differential value between two successive pixels [11]. In addition, the image interpolation-based scheme has shown excellent embedding capacity without affecting image quality [12].

There are quantum schemes included in this paper that have shown diverse and novel approaches for quantum steganography, with some being primarily innovated for quantum applications. An example is Nagy et al. [13]'s new approach to quantum steganography by manipulating the characteristics of the communication channel of an entangled pair of photons. In addition, Sharma et al. [14] implemented quantum Hilbert scrambling in secret images and embedded them into an image by the alpha blending operation. Other researchers have developed quantum renditions of classical works to create new approaches to steganography. For example, Yang et al. [15]'s use of the turtle-shell-based matrix, Luo et al. [9]'s use of inverted quantum images to produce a higher image quality, Sun et al. [16]'s newest approach of using classical protocols to develop the double-layer matrix coding protocol. In addition, Qu et al. [17]'s enhanced version of the exploiting modification direction protocol (EMD) [18] implements steganography in quantum contexts. The paper also discusses the four types of quantum noise channels, showing their effect on implementing steganography [19]. The structure of this survey paper is sketched in Figure 1, below.

In this survey paper, we provide the necessary background and components found in the quantum image steganography field, including the quantum image representations commonly used for data hiding. Moreover, we discuss the available solutions for compensating for the lack of real quantum computers for the implementation process, and the use of high-performance computing (HPC for short) to simulate quantum machines. The main goal behind this paper is to present the latest advances in implementing quantum steganography and image steganography along with the recent improvements implemented in these fields. Furthermore, we explore the most recent methodologies proposed as well as a brief segment in each methodology explaining its background history and the inspiration behind it. The paper also covers the results of each method discussed and the improvement each has created compared to previous research papers.

Our motivation behind conducting this survey is the absence of research papers summarizing the latest progress in quantum image steganography and the overall shortage of works presented in the quantum computing domain. In addition, we aim to encourage future research in quantum steganography, data hiding, and other related fields, by summarizing the most recent works presented in this domain and discussing the latest challenges faced in this domain. Additionally, our goal is to provide us and researchers interested in the HPC and quantum computing fields with an adequate source of information on all the necessary background to generate improvement in the quantum steganography field and quantum computing in general.

The survey paper includes five sections. In Section 2, we provide the necessary background by describing quantum image representation, while a detailed analysis and comparison of various schemes are shown in Section 3. In Section 4, we discuss the points made in the previous sections as well as the future opportunities for researchers in quantum steganography. In addition, lastly, we conclude our paper in Section 5.

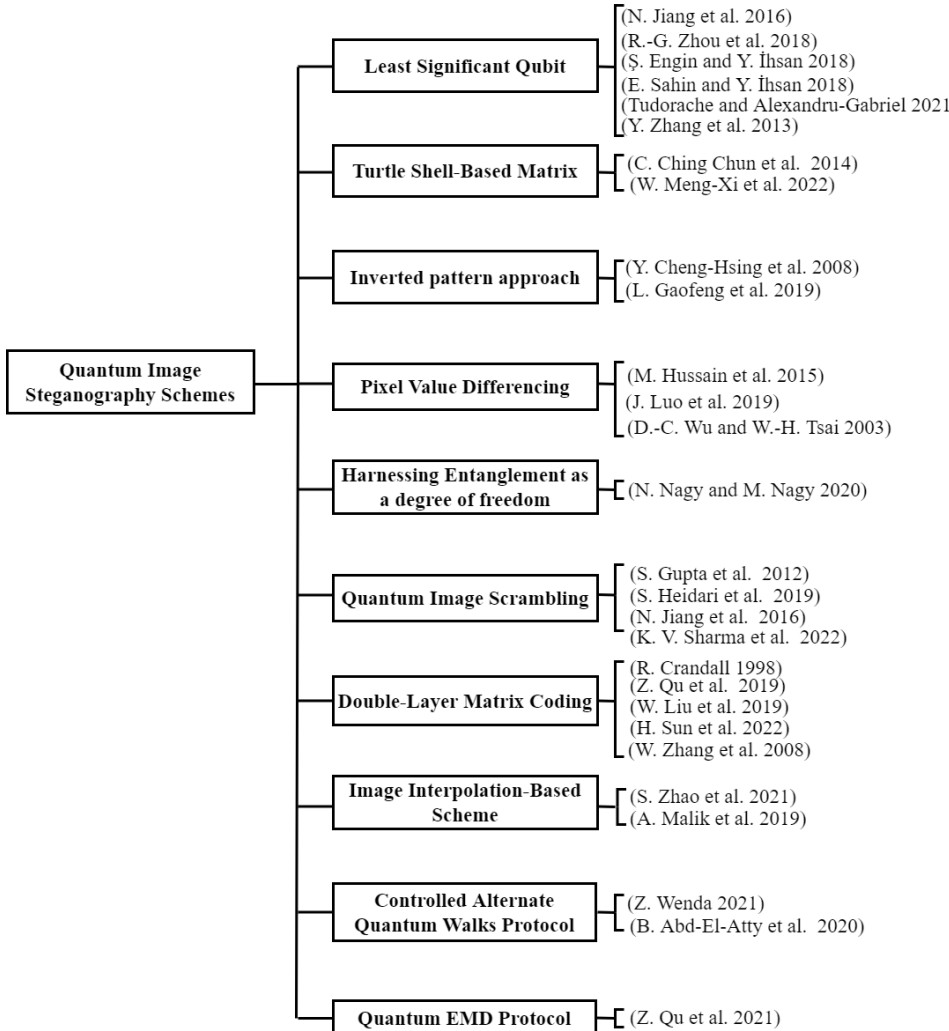

**Figure 1.** Recently implemented quantum image steganography schemes.

## 2. Background

Today, quantum computers provide a new paradigm for solving problems that are computationally intensive using traditional programming methods. However, before real quantum computers become mainstream, classical emulation with high-performance computing is an acceptable approach for the efficient simulation and simplification of quantum algorithms. For practical scenarios, quantum processing units (or QPUs) are expected to be added as accelerators at the hardware level for improved performance. For instance, QPUs can work together with graphics processing units (GPUs) in classic supercomputers. For this matter, a few hybrid algorithms have been proposed such as variational quantum eigensolver, variational factoring, etc. [20,21].

The concept of steganography has gone through a long history, as the first documented implementation was around 440 BC [22]. After that, steganography has been expressed through different mediums. From hiding messages using traditional communication methods and letters to using computers by hiding messages through classical bits and the employment of quantum information and protocols to implement steganography.

In steganography, attackers can detect and gain access to hidden data through a steganographic source if the scheme is not secure enough. For instance, if the embedding scheme is too simple to detect, it becomes easy for attackers to access the embedded information. Moreover, the hidden data must not be detectable when embedded in an image, making it important to increase the imperceptibility of a steganographic picture [23]. There are several attributes that a steganography scheme must include to succeed in achieving high security. The authors in [6,24] described a few main categories for calculating an image steganography's performance, which include:

- Hiding capacity—Evaluates the size of data that the scheme is capable to embed into the carrier image. The sizes can vary from hiding text segments to images equal to the size of the cover image.
- Imperceptibility—demonstrates the opaqueness of the hidden image and assesses whether the human eye can depict a steganographic image.
- Security—determines the difficulty for unauthorized parties to access and manipulate the hidden data.
- Robustness—exhibits how robust the hidden data is against received attacks and its ability to maintain its state.
- Unambiguity—the ability to identify the copyright owner of the secret data.

To implement quantum image steganography, researchers experimented with hiding images and messages with different quantum image representations [25]. With the presence of different models of quantum images, finding the appropriate image model for implementing a steganography scheme is essential, as each image representation comprises unique characteristics that distinguish it from other models [26]. For this reason, deciding on the appropriate type of image representation to be used in steganography is a crucial step to ensure the success of data hiding. The most popular image representations, namely, flexible representation of quantum images, novel enhanced quantum representation, new quantum representation of color digital images, and quantum multi-wavelength image representation are presented below [27]:

Flexible representation of quantum images (FRQI) is a model that produces information regarding an image's color as well as its position in a quantum state directly from a classical image representation, producing a grayscale image. The FRQI model can be represented using the following formula, as mentioned in [23]:

$$|I\rangle = \frac{1}{2^n} \sum_{i=0}^{2^{2n}-1} |c_i\rangle \otimes |i\rangle, \tag{1}$$

In the previous formula, $\otimes$ defines the tensor product notation, while $|c_i\rangle$ portrays the two-dimensional quantum states, as well as the image's color information, which corresponds to the angle vector represented by $\theta$. The value of $|c_i\rangle$ is portrayed in [23] by the following formula:

$$|c_i\rangle = \cos\theta_i |0\rangle + \sin\theta_i |1\rangle. \tag{2}$$

Novel enhanced quantum representation (NEQR) is an enhanced representation of the FRQI model, with the addition of storing the grayscale color information of each pixel in an image along with its position in the two entangled quantum sequences. The NEQR model for an image size $2^n \times 2^n$ are represented by the formula taken from [7]:

$$|I\rangle = \frac{1}{2^n} \sum_{y=0}^{2^n-1} \sum_{x=0}^{2^n-1} |f(y, x)\rangle|yx\rangle = \frac{1}{2^n} \sum_{y=0}^{2^n-1} \sum_{x=0}^{2^n-1} \overset{q-1}{\underset{x=0}{\otimes}} |C_{yx}^i\rangle |yx\rangle. \tag{3}$$

Given that $|C_{yx}^i\rangle$ represents the qubit sequence in a pixel, and $|f(y, x)\rangle$ demonstrates the grayscale range for an image.

Multi-channel representation for quantum images (MCQI) is an image model that uses a normalized state to store the RGB data of an image. Figure 2 represents the MCQI model and the formula that signifies it.

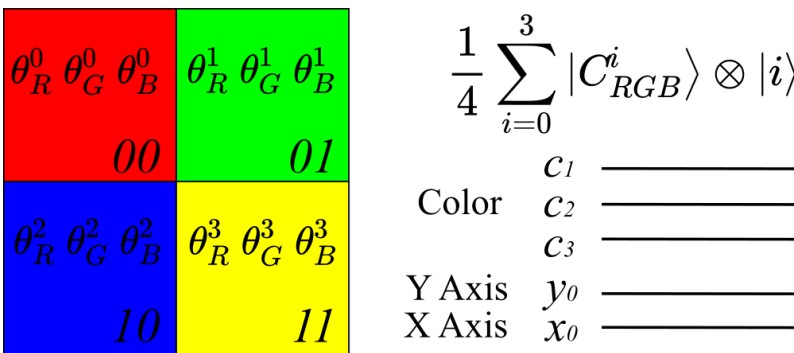

**Figure 2.** A simple $2 \times 2$ MCQI image representation. Adapted from [28].

New quantum representation of color digital images (QRCI) is a model which was directly inspired by NCQI, with the addition of using both RGB color information and bitplane to represent a color image. The color information for the three RGB channels is represented in [29] as:

$$C_L(Y, X) = R_{LYX}G_{LYX}B_{LYX.}  \tag{4}$$

Quantum multi-wavelength image representation (QRMW) is an image representation model that specializes in storing the image's wavelength, position, and color value for each pixel in three separate qubit states. The formula that represents this scheme, as stated in [8], is:

$$| I \rangle = \frac{1}{\sqrt{2^{b+n+m}}} \sum_{\lambda=0}^{2^b-1} \sum_{y=0}^{2^n-1} \sum_{x=0}^{2^m-1} | f(\lambda, y, x) \rangle \otimes | \lambda \rangle \otimes | yx \rangle.  \tag{5}$$

## 3. Quantum Image Steganography Schemes

In this section, we discuss ten quantum steganography techniques, highlight their pros and cons, and recommend future research directions, where applicable.

### 3.1. Least Significant Qubit

The least significant qubit (LSBq) is one of the most common techniques for applying quantum image steganography [25]. Inspired by the LSB method, LSBq explains the process of hiding data by substituting the least significant qubits of an image using the qubits of the confidential data [4]. Studies used LSB and LSBq methods to implement quantum steganography, with each showing different approaches to enhance this method by applying additional protocols and algorithms to improve the steganography representation and security. However, there are also recent studies using primarily the LSBq scheme to create novel quantum steganography approaches.

For instance, in 2018, Sahin et al. [5] achieved high security and efficiency by primarily using the LSBq approach. In this scheme, the embedded qubits were placed into the corresponding channel by a specified modulo value rather than in order, which, as a result, enhanced the steganography's security. Moreover, Sahin reduced the time complexity of the algorithm by directly embedding the covert message's qubits into the least significant qubit of the carrier image instead of comparing them as a step before exchanging them. The scheme used quantum multi-wavelength images (QRMW) [9] to implement steganography, which allowed for adding different wavelengths for each modulo value to hide larger scales of the secret message and increasing the scheme's security. By implementing the previous changes, the mentioned method presented a better performance compared to older works from the same domain.

A more recent approach by Gabriel et al. [6] created steganographic images using LSB by embedding the confidential image qubits as well as each of the two threshold values into the 16 pixels' least significant bits found at the edges of a grayscale image [7], where the same two thresholds are used to extract the embedded message by filtering out the cover image. Gabriel used the image model NEQR [7] for their scheme, which specializes in using qubits to hold information of grayscale values as well as each qubit's position. With the use of the NEQR model [7] and a novel LSB embedding scheme [6], the results of this method have shown a large increase in performance speed compared to the alternatives. Moreover, to improve this method, Gabriel suggested that the existence of an operation that resets the qubit in the future could further enhance its process.

### 3.2. Turtle Shell-Based Matrix

Chang et al. [24] were the first to apply the idea of embedding information using a turtle-shell-based matrix, which constitutes a numerical pattern that consists of turtle shells represented as hexagon shapes containing octal numbers from 0 to 7.

Inspired by Chang et al., Yang et al. [15] in 2022 implemented quantum steganography by applying the turtle shell matrix with LSB to hide an NEQR grayscale image into an NCQI color carrier image. The research by Yang et al. was the first to use a turtle-shell-based matrix as an embedding scheme for quantum steganography as well as simultaneously implementing other functionalities such as human vision system model (HVS), and LSB into their algorithm.

To choose the locations for hiding qubits, the algorithm selects various pairs of the cover image's qubits from three RGB color channels (red, green, and blue), where each has a different probability to be chosen depending on how sensitive the HVS is to it. This way, the algorithm will create either (blue, red) or (blue, green) qubit pairs as a guide for hiding. To increase the security and extend the image representation, the algorithm added an error detection code called a parity bit at the end of each binary sequence. After that, it hid three qubits, two in the blue channel LSB and one in either green or red channel LSB, using the turtle-shell-based matrix.

In the first step of the turtle-shell-based matrix scheme, the algorithm divides the carrier image into blocks, where each image block contains an entangled quantum state representing the coordination of each pixel. Then, based on the RGB qubit pair's grayscale value, the qubits will be embedded into the corresponding image blocks following an algorithm that dictates their position. Based on the results from comparing the proposed turtle-shell-based matrix approach's performance to other papers, the method has displayed a significant improvement in security, and achieves better trade-off between image quality and embedding capacity compared to older schemes [15].

### 3.3. Inverted Pattern Approach

Inspired by the classical inverted pattern approach by Yang et al. [30], recent research for applying quantum steganography aimed to achieve excellent performances in embedding capacity and visual effects while simultaneously achieving a low computational complexity by using a quantum version of the inverted pattern scheme along with LSB [9]. The benefit of applying the inverted pattern approach [30] to a scheme is to improve the steganographic image's resolution after being degraded from applying the LSB method.

In this scheme, Luo et al. [9] first used LSB to hide the secret message content into an NQER image model by embedding each qubit into each image pixel's least significant bit. The algorithm then applies the inverted pattern approach by transforming some of the secret message pixels into an inverted qubit sequence, where it records the produced inverted pattern using an empty quantum binary image. Next, to improve the steganographic image quality, the algorithm compares the original embedded image with the inverted one and uses the most appropriate sequence out of the two to create the ideal qubit sequence and produce a better image. When extracting the secret message, the algorithm inverts the embedding process and adds it to an empty quantum image.

Along with achieving a lower computational complexity than other quantum steganography schemes and its superiority in visual quality and capacity, Luo's approach of using the inverted pattern approach to enhance LSB is straightforward to execute.

### 3.4. Pixel Value Differencing

A good steganographic algorithm, pixel value differencing (PVD), is known for its superior spatial visual recognition as well as its high payload [31]. The PVD technique determines how many hidden bits can be inserted by finding the subtraction of two successive pixels.

In 2019, Luo et al. [11] proposed a scheme for implementing quantum steganography by adopting the PVD approach to enhance the robustness and performance of his quantum steganography algorithm. The concept of PVD was initially proposed by Da-Chun et al. [32] in 2003; it was discovered that each pixel in a digital image may survive a different degree of alteration because the human eyes exhibit more sensitivity to any changes to pixels applied in the smooth section of an image rather than its edges. That is, every pixel in an image can contain a varied quantity of covert bits without creating detectable sensory distortion. However, with the LSB steganography technique, the amount of alteration in each pixel is uniform. As the LSB method alone does not provide an examination of each pixel and ignores the effect of using an image's edges, its performance is generic.

In Luo's approach [11], the PVD method first partitions a cover picture into blocks with each containing two successive pixels. Then, by subtracting the two pixels, the algorithm will extract the difference between them, signified by d. The scheme then takes the absolute value for every d, dividing them into several neighboring ranges known as Ri, where i represents a sequence of numbers from 1 to n. The total bits to be embedded in each pair of pixels are then determined by the corresponding range's extracted difference value. After that, the difference will be subsequently substituted with the secret information bits. This approach is a straightforward way to generate a more unnoticeable outcome than the simple LSB procedures. Furthermore, the covert data contained in the stego image may be recovered without referencing the original image. To achieve this method, a sequence of reversible logic circuits is designed.

### 3.5. Quantum Noisy Channels Effect

Quantum noise must be considered when it comes to communication because it could substantially damage a quantum communication system. Channel coding is a good way to counteract the impacts of noise on channels [33]. There are four main types of quantum noises that can occur in quantum channels:

#### 3.5.1. Amplitude Damping

This quantum noise channel refers to the process of energy being lost in a quantum system, also known as energy dissipation. The formulation of this quantum noise is shown in [19] below:

$$E_0 = \begin{pmatrix} 1 & 0 \\ 0 & \sqrt{1-p} \end{pmatrix}, \ E_1 = \begin{pmatrix} 0 & \sqrt{p} \\ 0 & 0 \end{pmatrix}, \tag{6}$$

The noise factor $p$ indicates the probability of losing a photon in a quantum system.

#### 3.5.2. Phase Damping Channel

The spin–spin relaxation process indicated as T2 (also known as phase damping) is defined as the process of interfering without the loss of energy to quantum information. The formula for this quantum noise represented in [19] can be written as:

$$E_0 = \sqrt{1-p}I, \ E_1 = \sqrt{p}. \tag{7}$$

The probability of a photon reflecting without the system losing energy is represented by the noise coefficient $p$, and the Pauli matrix is represented by the value of $\sigma_z$.

### 3.5.3. Bit Flip Channel

This is a quantum noise channel that flips the state of a qubit from $|\ \rangle$ to $|\ \rangle$ or from $|1\rangle$ to $|0\rangle$ with probability p. The quantum operation for the bit flip channel is represented in [19] as:

$$E_0 = \sqrt{1 - p}I, \ E_1 = \sqrt{p}\sigma_x, \tag{8}$$

### 3.5.4. Depolarizing Channel

This quantum noise causes a qubit to become a mixed state $I/2$ by depolarizing it with a probability of $p$. The operand for this noise channel is depicted in [19] as:

$$E_0 = \sqrt{1 - p}I, \ E_1 = \sqrt{\frac{P}{3}}\sigma_z, \ E_3 = \sqrt{\frac{P}{3}}\sigma_y, \tag{9}$$

where $I, \sigma_x, \sigma_y, \sigma_z$ are four Pauli matrices.

The steganographic protocol will be influenced by quantum channel noise when transmitting information and the impact of noise on quantum states can be described as fidelity $F$, the term "fidelity" is defined in [19] as follows:

$$F = \langle T|\rho_2|\ T\rangle. \tag{10}$$

Fidelity levels vary from 0 to 1. The similarity between the two quantum states increases with increasing fidelity. The two quantum states are identical if the fidelity is 1.

Findings from the experiment that test the performance of the steganographic protocol based on the $\chi$ state under quantum noise by Sun et al. [19] demonstrate that when just the sender's first transmission in a quantum channel is impacted by quantum noise, and the second unaffected by noise, the first protocol is least affected by amplitude damping noise. In addition when both the sender's first and second transmission channels are impacted by quantum noise, it is discovered that the efficiency of the protocol in a quantum noisy channel will be improved by a particular mix of several noises. As a result, the protocol can function better when affected by quantum noises by purposefully adding additional quantum noise to the quantum channel by the noise intensity.

### 3.6. Entanglement as a Degree of Freedom

Briefly, the concept suggested by Nagy et al. introduces a novel approach to quantum steganography which manipulates the characteristics of the communication channel of an entangled pair of photons [13]. The entangled duo's channel combines two domains, entanglement, and polarization. The intended exploiting of the medium occurs when the entanglement domain is used as a secondary method to represent the concealed message. Conversely, the cover data used as a disguise is encoded within the polarization domain.

First, for this quantum steganography scheme to be employed, it must distinguish between an entangled pair and an independent single photon. To detect entanglement an interferometer is used, with respect to a Mach–Zehnder interferometer in Figure 3, which is applied for analysis.

Figure 3 illustrates the regular composition of a Mach–Zehnder interferometer. The beam splitter BS is normally considered a semi-reflecting mirror, built using a transparent material with one side reflecting. The mentioned interferometer comprises two base vectors named lower arm and upper arm, which depict a photon's state spanning the interferometer either through its lower or upper arm, respectively.

$$|\ L\rangle = \begin{pmatrix} 1 \\ 0 \end{pmatrix}, |\ U\rangle - \begin{pmatrix} 0 \\ 1 \end{pmatrix}. \tag{11}$$

Due to the beam splitter being reflective on one side, the first beam splitter (BS1) will allow the reversing to the upper arm $U$ only for half of the incident beam, while the second half of the beam will go across the lower arm $L$. By quantum mechanics, the photon is

simultaneously mirrored through *U* and crosses into *L*. The below matrix expresses the behavior as a model of the quantum operator:

$$BS1 = \frac{1}{\sqrt{2}} \begin{pmatrix} e^{2i\theta} & e^{i\theta} \\ e^{i\theta} & -1 \end{pmatrix} \tag{12}$$

Secondly, after birthing the photon, the information can be encoded within the entanglement domain. Codes, for instance, 0's and 1's bits are determined by the physical process that generated that photon. Values of induced bits are known by the triggering of a certain photon detector in the interferometer. Specifically, when a photon is transferred via a Mach–Zehnder interferometer, it can provoke a 1 or 0 reading by correspondingly exiting the apparatus vertically or horizontally. A vertical exit results in a registered click by D2. However, a horizontal exit issues a click by D1. Often, entangled photons' way out are performed horizontally.

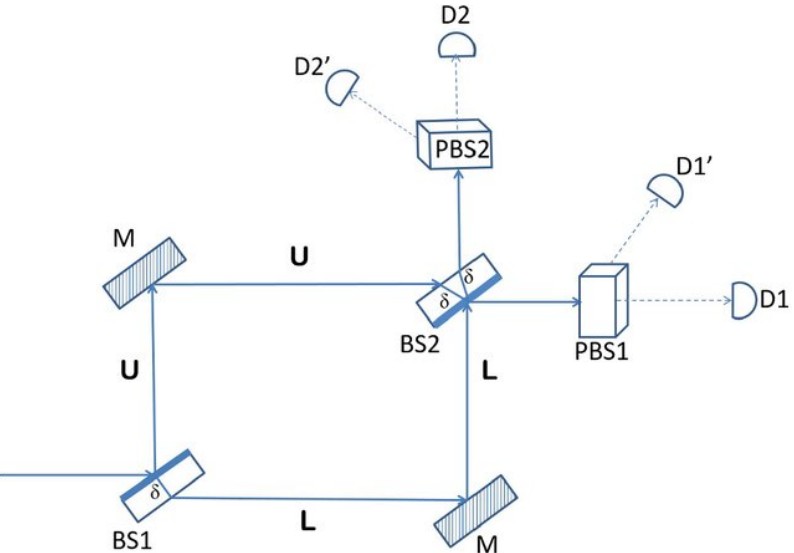

**Figure 3.** Mach–Zehnder interferometer ([13]) .

Finally, the scheme implements quantum steganography by concealing secret data with the host data. Both can hold any format (image, video, audio, text, etc.). The results of this scheme have shown an outstanding performance in all areas, such as embedding capacity, as it can hide an image equal to the size of the cover image, along with its ability to cause no degradation to the stego image's quality, and its simplicity and adaptability. Since the two states of photons are independent, both secret and host data are separate, such that any modifications made to the carrier image will not affect the hidden image.

### 3.7. Quantum Image Scrambling

Image steganography discusses the concept of disguising data inside of an image to transfer to the recipient, such that only the recipient can distinguish the existence of the secret data [34]. Quantum image scrambling is a pre-processing method that is applied through an image-scrambling algorithm which accounts for the quantum representations of the hidden image's data in a secure manner that turns the quantum image into a disordered image before sending it to the receiver [35]. As a result, unauthorized attackers would just be met with a meaningless image had they succeeded in the attempt of extracting the data.

The concept of quantum image scrambling was proposed in 2016 by Jiang et al. [3] who introduced their methodology using the image model FRQI by employing an Arnold and Fibonacci transition. Correspondingly, an alternative quantum image-scrambling method that utilizes an algorithm of the Hilbert scanning matrix called quantum Hilbert image scrambling was introduced to scramble a flexible representation of quantum images.

In 2022, this concept was incorporated into recent research conducted by Sharma et al. [14], where they utilized the disordered version of the covert image and an applied graph wavelet transformation on both the disordered covert image and the cover image while subsequently alpha blending both image signals during the embedding. Alpha blending $\alpha$ is the method of conjoining two images to produce a new one. Their research indicates that their image steganography method uses images as carrier media and hidden media. It also comprises two main frameworks, which are the implanting framework and the withdrawal framework of the covert image.

Using quantum scrambling, the implanting framework in the proposed scheme scrambles the hidden image. The quantum Hilbert image-scrambling protocol produces exceptional scrambled media by using layered as well as recursive circuits, establishing confidentiality and robustness. After that, a graph wavelet analysis is applied to both the scrambled image along with the carrier image. Lastly, the proposed scheme applies the alpha blending function on the produced graph wavelets of the scrambled confidential image with the cover image to achieve the stego image.

$$(Ab)_G = (Ci)_G + \alpha(SSi)_G \tag{13}$$

The $(Ab)_G$ refers to the stego image produced after the blending procedure. Meanwhile, $(Ci)_G$ denotes the cover image and $(SSi)_G$ represents the scrambled covert image. In the mentioned formula, the $\alpha$ indicates the embedding strength aspect for ruling the robustness of the produced image and its imperceptibility. A stego image is then obtained after the graph wavelet-synthesis process is applied to the image generated by the merging operation shown in Formula (13).

For the withdrawal framework stage, a graph wavelet-synthesis analysis is first operated on both the hidden and cover image. Afterward, the scheme attains the scrambled image $(SSi)_G$ by applying the alpha blending procedure to the produced images. Consequently, the procedure performs a graph wavelet-based synthesis operation on the steganographic image followed by the inversed Hilbert image scrambling protocol, attaining the hidden image.

### 3.8. Double-Layer Matrix Coding

Crandall et al. [36] were the first to propose the concept of matrix coding in 1998, which was the foundation of linear-codes-based steganography and numerous other classic steganography schemes inspired by his procedure. Its key notion is to utilize the Hamming code by implementing it into a steganography procedure to obtain great capacity and high embedding efficiency.

In association with linear-codes-based quantum steganography, Qu et al. [37] proposed a quantum image steganography protocol based on matrix coding. The scheme proposed by Qu was an expansion of the matrix coding steganography initially produced in the quantum steganography sphere, and it maintains the circuit design used to transmit secret information with respect to its corresponding notion about the quantum carrier image proposed by Qu et al.

To further advance the previously proposed scheme, research conducted by Sun et al. [16] in 2022 established two protocols inspired by the double-layer matrix coding approach. The first protocol introduced by Sun was derived from Zhang Weiming, Zhang Xinpeng, and Wang Shuozhong's method to implement steganography (ZZW) [38]. Furthermore, the ZZW framework originated by Zhang et al. [38] is considered a double-layer matrix coding protocol which has two main elements in its procedure, namely, the hamming code and the wet-paper code. In the proposed framework, the pixel blocks can hide the covert data through the two channels utilized for embedding with one alteration. By applying this approach with the use of a quantum carrier image, Sun's scheme achieved both high security and enhanced embedding efficiency.

The second protocol proposes an enhanced version of the quantum image steganography protocol that holds a capacity twice as large as the first protocol and achieves an even

higher embedding efficiency. To exhibit the benefits of using both protocols, Sun et al. [16] designed specialized quantum circuits that implement both of the previously introduced frameworks to hide and withdraw covert data, proving that implementing both protocols together results in achieving high imperceptibility along with great embedding efficiency.

### 3.9. Interpolation-Based Scheme

In quantum steganography, the cost of aiming to have a high embedding capacity is a lower image resolution. For this reason, researchers tried to find diverse ways to regain their image quality. One of the solutions provided by researchers is to employ an interpolation method into steganography and data hiding schemes [39], which is a method for producing high-resolution images from lower ones by resizing them. The previous approaches focused primarily on applying interpolation-based algorithms using classical computing. A recent approach by Zhao et al. [12] implemented quantum steganography by embedding confidential messages into a NEQR model image using image interpolation [40].

Before embedding, the algorithm increases the cover image's size using quantum image interpolation, producing an up-scaled version of it. The scheme then calculates the interpolation error by calculating the difference between the up-scaled image and the former one. Next, the algorithm will calculate the number of qubits to be hidden in each pixel using the formula, which represents the interpolation errors. Then, based on the calculated capacity for each pixel, the confidential message qubits will be hidden accordingly. Each pixel is edited at least once after embedding to produce a steganographic image. The confidential image is later recovered by extracting the initial pixels and deleting the inserted ones using quantum nearest neighbor interpolation.

The results of following this method have shown an excellent embedding capacity without affecting the image quality compared to relative schemes. Moreover, Zhao et al. [12] suggested that with more improvement, the approach could deliver better results, especially for the trade-off between capacity and invisibility.

### 3.10. Controlled Alternate Quantum Walks Protocol

Inspired by random walks, a newly designed quantum protocol by Wenda et al. named controlled alternate quantum walks (CAQWs) [41] can outperform classical computers in terms of exponential algorithmic speedup by taking advantage of the quantum state characteristics.

Based on CAQWs, Abd-El-Atty et al. [10] promoted protected transmission in cloud-based E-healthcare systems through the development of an innovative steganography scheme. The primary task of CAQWs is to select two LSB pixel places in the carrier picture for the hidden bits to be embedded, which makes the outcome capacity of the algorithm two bits per pixel.

Before the embedding process, the algorithm transforms the carrier image, detection matrix, and private image information into a vector using CAQWs. Additionally, CAQWs will carry out the vectorization process on a series of odd-valued nodes specified by a bit string using the preliminary parameters, yielding a probability distribution matrix.

The proposed steganography approach removes some requirements such as the before and after encryption alongside the extraction procedures, which means that only stego images and main states of the CAQWs are obligatory to extract hidden images, making the process easier. The plan performs several processes that include extraction and embedding according to the specifications of CAQWs, and those processes are then utilized to decide upon pixels for the embedded hidden bits in the stego picture.

Multiple researchers have gone through a lot of experiments and trials to establish an efficient scheme capable of processing a collection of medical photos in color and grayscale regarding good visual quality, resistance to data loss hazards, as well as making sure it meets high embedding capacity and meeting all those goals while having strong security. These experiments had several outcomes that prove that the quantum-inspired scheme achieves satisfactory results compared to similar techniques; they also proposed

some possible applications that are qualified for conveying successful medical image steganography on upcoming computing models [10].

### *3.11. Quantum EMD Protocol*

The exploiting modification direction steganography protocol (EMD) is a classical scheme presented by Zhang et al. [18] which inspired future quantum renditions. The method operates by converting a hidden text, image, or any type of message into a series of digits with an odd base in the notational system, then all digits belonging to the $(2n + 1)$-ary from the system are moved by $n$ pixels. In addition, these pixels can be added or subtracted by 1.

To apply this method into quantum image representations, Qu et al. [42] created a quantum version of the EMD steganography protocol (QIS-EMD). The image model used for this process was NEQR. The concept of EMD embedding suggests that $n$ pixels will capture each Dt belonging to the $(2n + 1)$ carrying system. Below, the listed steps (with $N = 2$) represent the embedding and extraction processes done by the new algorithm:

To store the data of gray value gn ($0 \leq$ gn $\leq 255$) of the digital gray image, it requires 8 qubits in NEQR per 1 pixel. For instance, a quantum carrier image $\mid C\rangle$ having $0 \leq$ gn $\leq 255$ and size $2^n \times 2^n$ is consequently indicated as:

$$\mid C\rangle = \frac{1}{2^n}\sum_{i=0}^{2^{2n}-1} \mid c(i)\rangle \mid i\rangle = \frac{1}{2^n}\sum_{i=0}^{2^{2n}-1} \mid c_7^i c_6^i \ldots c_0^i\rangle \mid i\rangle \tag{14}$$

On the sender's end, with each comprising of $L$ qubits, the binary flow of hidden information is segmented into a vast number of parts. The decimal digit of each hidden fragment is equal to:

$$L = \lceil P \bullet \log_2 5 \rceil \tag{15}$$

where $P$ is the digits included in the above-mentioned notational system. With EMD embedding, each Dt maps to a comparable carrier pixel-group. Once all are embedded, the quantum carrier image $\mid C\rangle$ is converted to a quantum stego image $\mid C'\rangle$. This completes the embedding section.

The extraction function is applied by the receiver on the stego pixel group to draw out an equivalent secret digit. To complete the extraction section, each drawn-out secret digit is converted into a secret piece, building up the full binary stream of hidden information.

In 2021, Qu et al. [17] further enhanced the QIS-EMD scheme by creating the improved quantum EMD protocol (QIS-IEMD), where two additional improvements were added to the previous scheme. The first is employing the expanding modification range method, which modifies the carrier pixel's values of numerous bit planes to achieve good imperceptibility and enlarge the capacity to its greatest extent. After the employment of this modern protocol, an addition or subtraction by 1 can be applied on the modified pixels, signifying the level of modification depending on the modification range. Moreover, $L$'s upper bound increases, ensuing a new value of $L$ that equals:

$$L = \lfloor \log_2 (2cn + 1) \rfloor \tag{16}$$

The second protocol added to the new variation is the dynamic sharing of subgroups, which aims to achieve better embedding efficiency and compress the redundancy by making neighboring pixel groups be shared dynamically together. As such, during the embedding procedure, after modifying one pixel possessed by the 1st pixel group, the next pixel group will start the modification, sharing the rest of the pixels with the modified one in the first group. However, it must be noted that the modification of pixels shared between the allocated groups is prohibited.

Finally, we provide a summary of all the recently presented quantum steganography schemes in Table 1. Moreover, we included a short description of each scheme's methodology, along with the latest advances and outcomes.

**Table 1.** Summary of the Latest Quantum Steganography Schemes and Their Outcomes.

| Scheme | Methodology | Recent Outcomes | Limitations |
|---|---|---|---|
| LSBq [4,6,8] | Embeds the secret message's qubits to the carrier image's least significant qubits. | • Researchers increased LSB's security by embedding it by a modulo value rather than in order.<br>• Enhanced the security of LSB by using thresholds values.<br>• Implemented QRMW image model to increase the LSB method's capacity. | Low robustness: Using primarily LSBq makes it easy for intruders to detect and extract the steganographic image. Using the approach solely can result in low imperceptibility. |
| Turtle Shell-Based Matrix [15] | Embeds the secret message's qubits into an image based on a turtle-shell pattern. | • Embedded qubits using diverse embedding approaches along with the turtle shell matrix which increased its security.<br>• Achieved high embedding capacity, reaching 3 bits per pixel.<br>• Used HVS's varying sensitivity to the three RGB pixels to detect which RGB pair the qubits should be embedded in each pixel, increasing the quality of the image. | The proposed scheme has only been experimented on using a simulated environment and not on a real quantum computer. |
| Inverted Pattern Approach [9] | Compares two versions of a stego image (inverted and original) then chooses the best sequence out of the two to produce an optimized version of the image. | • Used IPA to enhance the stego image's resolution after the embedding procedure.<br>• Achieved higher image quality with a lower computational complexity.<br>• Provided high embedding capacity. | The scheme achieved low robustness since it primarily uses LSB to embed secret data. |
| PVD [11] | Originated from its classical counterpart, which finds the subtraction of two successive pixels to detect the number of bits to be embedded in each pixel. | • Implemented PVD in quantum contexts to generate a straightforward and more unnoticeable steganography than the simple LSB procedure.<br>• The covert data contained in the quantum stego image can be recovered without referencing the original image. | Only less sensitive data can be concealed when there is little pixel difference inside the pixel block. |
| Quantum Noisy Channels Effect [19] | Exploits quantum noisy channels' effect on qubits in the transmitting stage. | • It concluded that a steganography protocol can function better when affected by quantum noises by purposefully adding additional quantum noise to the quantum channel by the noise intensity. | Could substantially damage a quantum communication system. |
| Entanglement as a Degree of Freedom [13] | A novel approach which manipulates the features of a quantum communication channel for an entangled pair of photons | • Implemented a new approach to steganography by using the depolarization domain and entanglement domain to hide secret images into a carrier image.<br>• The scheme created achieved exceptional capacity levels as well as high robustness due to the independent nature of the entangled photons. | Difficult to implement since it requires advanced equipment and tools to be able to distinguish singular photons from entangled ones. |

**Table 1.** *Cont.*

| Scheme | Methodology | Recent Outcomes | Limitations |
|---|---|---|---|
| Quantum Image Scrambling [14] | Scrambles a piece of quantum media using the Hilbert scrambling protocol to produce a hidden disordered outcome before sending it to the receiver. | • Applied quantum scrambling protocol into the hidden image prior to the embedding process to increase security and confidentiality.<br>• Used alpha blending to embed the disordered image into the cover image to achieve robustness and increase the scheme's security. | It is more robust and offers less of a bit error rate than other techniques, but its channels could still be affected by noise during data transmission. |
| Double-Layer Matrix Coding [16] | Based on the original matrix coding approach which employs the Hamming code and wet-paper code to achieve steganography. | • Created two advanced protocols based on the double-layer matrix coding approach, which both achieved superior embedding capacity and efficiency. | As it utilizes the idea of wet paper code, it only modifies a selective number of pixel blocks and not all of them after embedding secret information in the first embedding channel. |
| Interpolation-Based Scheme [12] | Upscales the cover image using the interpolation algorithm and finding the difference between the new and original image to decide the number of bits to be embedded in each pixel. | • Produced steganographic images with a superior image quality.<br>• Attained high embedding capacity without degrading the image's resolution. | The algorithm used can be perceived as high in time complexity, making it slower in performance compared to other schemes. |
| CAQWs [10] | A quantum alternative of the random walks protocol to choose locations in a random manner through a probability distribution matrix. | • Used CAQWs protocol to detect LSB embedding locations to hide quantum images into a cover image, achieving strong security and high resistance to data hazards. | In some situations, processing a huge collection of images with good visual quality, resistance to data loss hazards, and high embedding capacity might not be granted |
| EMD [17] | Hides a piece of media into a cover by embedding each qubit using the odd base carrying system. | • Created a quantum version of the EMD scheme to apply it into NEQR image representation.<br>• Enhanced the quantum EMD approach by expanding the modification range of the carrier image pixels to produce better imperceptibility.<br>• Applied the dynamic sharing protocol into the scheme to improve the embedding efficiency. | The embedding rate and capacity gradually decline as the group size increases, so there is a limit for how large it can become. |

## 4. Discussion

Thus far, we have explored ten of the recently proposed schemes for applying quantum image steganography. Among all the schemes discussed in this paper, there are two main domains that exist in the classical and quantum steganography fields. The first is the spatial domain, which explains using data components to embed the secret data in it, and in this case, the image's pixels were used as a hiding place for embedding secret images [43]. While in the transform domain, the image is converted from the spatial to frequency domain, embedding the secret data into the image's wavelets [44].

As mentioned above, based on both the image's nature and hosting places, the techniques of image steganography encoding are grouped into two popular divisions, one of which is the spatial domain. This class (also known as the map domain) uses bit insertion and bit-wise operation to hide data directly when converting an image's pixel values[45].

It is considered popular due to the advantages it offers. For instance, the spatial domain grants lower computational complexity than the transform domain. Additionally, higher imperceptibility and embedding capacity are also perceived in the spatial domain. In our survey paper, multiple approaches following the map domain class were discussed including the least significant qubit, turtle-shell-based matrix, inverted pattern approach, pixel value differencing, EMD, and interpolation-based scheme.

The spatial domain has included many representations and has shown improvements in quantum image steganography, making it a popular domain in the quantum-steganography field. An example is Sahin et al. [6], who used the least significant qubit technique that resulted in a reduction in the time complexity of the algorithm by directly embedding the covert message's qubits into the least significant qubit of the carrier image. Moreover, as the experiments of Gabriel et al. [6] showed, using the NEQR model [7] and a novel LSB embedding scheme, results in a large increase in performance speed. Chang et al. [15] used the turtle-shell-based matrix technique which showed an increase in security by utilizing an algorithm that added an error detection code called a parity bit at the end of each binary sequence, ultimately achieving a better trade off between image quality and embedding capacity.

In the spatial domain, the least significant qubit (LSBq) is considered the most popular out of the rest of the approaches and is widely known for its easy implementation and low time complexity [45]. However, it can also be deemed as the weakest in performance compared to others due to its low robustness and security. For this reason, we have seen multiple authors such as Sahin and Gabriel et al. improve the scheme by applying changes to its structure and adding elements that could improve the scheme's vulnerable spots. We have also seen many approaches that use LSB and LSBq in their algorithm but only as a supporting part. For instance, the inverted pattern approach by Luo et al. used the inverted pattern scheme along with LSB to increase the image's quality and imperceptibility by comparing each pixel of the embedded image along with an inverted version of it to find the best sequence out of the two and use it to produce the final image. In addition to that, there were improvements shown in the inverted pattern approach technique by Luo et al. [9], who were the first to use LSB to hide the secret message content into an NQER image model and achieve a lower computational complexity and superiority in visual quality and capacity.

We have also explored the pixel value differencing approach used in quantum contexts, which is also considered as a type of spatial domain. Luo et al. [9] accomplished superior spatial visual recognition as well as high payload [31] using the pixel value differencing technique that utilizes a good steganographic algorithm. Furthermore, Zhao et al. [12] found a way to reach an excellent embedding capacity without affecting the image quality with an approach in which embedding confidential messages into an NEQR model image was performed using image interpolation; the approach is known as the interpolation-based scheme. One of the disadvantages of using the spacial domain is its vulnerability to attacks, as it does not have as much security as the transform domain.

For the transform domain, an advantage of it is its high protection against compression and geometric assaults. However, the disadvantages shown in the transform domain are its high complexity, limitation of space for embedding images, and its overall lower perceptible controllability. One of the approaches presented in this paper that relates to the transform domain is Abd-El-Atty et al. [10]'s scheme, which has reached good visual quality, resistance to data loss hazards, as well as high embedding capacity, and strong security by using the controlled alternate quantum walks protocol. The way Abd-El-Atty's scheme utilized the transform domain is by converting the cover images into wavelet transformation before embedding the secret data, which the author then applied to a probability distribution matrix to detect the embedding locations for hiding. Another example of transform domain applications is Sharma's approach of implementing Hilbert image scrambling, where information was embedded into a quantum image representation by utilizing the wavelets of the cover image. The author also significantly increased the security of this scheme by

using the alpha blending operation into the image's wavelets, thus taking advantage of the wavelet transformation attributes for the images.

To encourage novel future work on the means of the spatial domain's embedding image steganography, a few open issues must be addressed to motivate more efforts from researchers, such as the security in LSB and EMD algorithms as well as LSB and PVD robustness. Moreover, researchers are encouraged to explore the transform domain's applications in quantum steganography, as, despite its high popularity among researchers, the domain still needs further work when applied to quantum contexts.

## 5. Conclusions

In conclusion, this survey paper encompasses a discussion on the field of quantum steganography, specifically examining five sections from demonstrating the background of quantum image representation to interpreting and evaluating diverse methods for achieving image steganography. Approaches reviewed are listed below:

1. Least significant qubit: uses the carrier image's least significant qubits to embed the needed secret qubits.
2. Turtle-shell-based matrix: secret message's qubits are encoded following a turtle-shell pattern.
3. Inverted pattern approach: compares and chooses the best sequence between the inverted and original stego image to generate a new advanced version.
4. Pixel value differencing: detects the appropriate number of bits to embed in each pixel by subtracting two successive pixels.
5. Quantum noisy channels effects: manipulates the effects of these channels on the bits during transfer.
6. Entanglement as a degree of freedom: concerned with photons as an entangled pair and utilizes quantum communication channels for it.
7. Quantum image scrambling: applies Hilbert scrambling protocol on a quantum media's fragment resulting in a hidden disordered output that is then transported to the receiver.
8. Double-layer matrix coding: attains steganography by employing both Hamming and paper codes.
9. Interpolation-sased scheme: determines the number of bits to embed in each pixel by comparing differences between the upscaled and original images after using the interpolation algorithm.
10. Controlled alternate quantum walks protocol: considered as random walks protocol's quantum substitute; a random location is chosen by a probability distribution matrix.
11. Quantum EMD protocol: implements the odd base carrying eystem to obtain embedding each media's piece's qubit into a cover.

One of the many limitations showcased in authors' works is the use of spatial and transform domains, since although the spatial domain offers an easy implementation, it still demonstrated a low rate of robustness. Conversely, the transform domains exhibited a high complexity and a limited image embedding space despite its minimal bit error rate and robustness. Another limitation is that a lot of the schemes were implemented using simulations instead of real quantum computers due to their general inaccessibility to the public. Furthermore, one of the inconveniences that we faced while writing this paper is that there were not enough research papers in the quantum steganography branch since it is still considered an inchoate field. Moreover, the concept of quantum computing is not yet available to the general public. Due to that, we faced plenty of difficulties in becoming familiar with quantum-related concepts.

On the other hand, diverse future work ideas were suggested by multiple researchers to enhance the concept of image steganography. For example, Gabriel advised improving the least significant qubit method by including a reset operation for the qubit. Additionally, to convey outstanding image steganography in the medical sector, researchers working with the controlled alternate quantum walks protocol proposed possible applications to

work on modern computing models. For our future work, and as authors and researchers in the field, we are starting our steps towards integrating the concept of oblivious transfer in the process of attaining quantum image steganography, which combines cryptography using various protocols along with image steganography schemes in the process of data embedding, thus, increasing the security of hidden information.

**Author Contributions:** Writing—original draft, M.A. (Mariam Alkharraa), M.A. (Mashael Alqahtani), D.A., R.S. and R.A.; Writing—review & editing, N.M.-A., N.N. and M.A. (Malak Aljabri). All authors have read and agreed to the published version of the manuscript.

**Funding:** The authors would like to thank SAUDI ARAMCO Cybersecurity Chair, Imam Abdulrahman Bin Faisal University for funding this project.

**Institutional Review Board Statement:** Not applicable.

**Informed Consent Statement:** Not applicable.

**Data Availability Statement:** Not applicable.

**Acknowledgments:** Authors would like to thank anonymous reviewers and for the excellent feedback on initial versions of the paper. Authors also extend their thanks to the Department of Computer Science, College of Computer Science and Information Technology, Imam Abdulrahman Bin Faisal University for facilitating this work.

**Conflicts of Interest:** The authors declare no conflict of interest.

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
