# Peer review of "Quantum Image Steganography Schemes for Data Hiding: A Survey"

_applsci, doi:10.3390/app122010294_

Round 1

Reviewer 1 Report

The paper titled “Recent Quantum Image Steganography Schemes for Data Hiding: A Survey” have shown the latest implementations for quantum steganography and described the history of each and how every paper reached into its current approach.

The paper discussed each method well and showed the features of each approach their strength points, and the solutions provided for various problems faced when creating steganography schemes. It also did a good job at explaining unfamiliar acronyms in the quantum steganography field. A summary table was shown at the end of the page for all the discussed schemes, which helps in understanding these approaches and makes it easier to observe the differences between them.

However, there are few points that I’d like to note in this paper.

First, I think it would be better if we increase the number of figures that illustrates the paper’s overall structure.

 Additionally, the ordering presented in the third section of the paper is unclear, so it would be more efficient if the schemes order followed a certain structure.

 The conclusion should be revised.

The image interpolation approach in this paper describes the idea of upscaling an image to embed another image through the difference between the original and the upscaled image. However, it not knows how they differentiate. I think it would be more suitable to add a brief background description for this approach.

Overall, the paper did a great job at delivering its goals. The description of each scheme is well written and straightforward, the short background provided in the schemes made it easy to understand them.

Author Response

As authors, we deeply appreciate taking the time to read and evaluate our work and for the brilliant review. And we are very thankful for your kind suggestions. Hopefully the updated version lives up to the required standards. Our rebuttal is elaborated below:

  1. We appreciate your suggestion, and we agree that increasing it will make the manuscript more understandable. The goal of this survey paper is to be straightforward and easily interpreted by the reader. Thus, the figures were improved to supply better understanding of each section. However, we concluded that the number of figures represented in this paper was sufficient, as the paper’s structure was simple and increasing the figures might cause distractions.
  2. Thank you for the observation. We have checked the ordering in section 3, and the reason why we haven’t followed a certain ordering scheme was because the topics were mostly independent from each other. However, we made LSBq to be the first topic to be discussed since many other schemes incorporate elements of this approach.
  3. We’re thankful for your suggestion. The conclusion was adjusted, and important points were highlighted.
  4. We appreciate the kind observation. Following your suggestion, have provided a brief background of the interpolation-based approach to provide a further understanding into this method.

Reviewer 2 Report

Some updates are required in the paper.

1.     There is no need for “Recent” in the title of the paper.

2.     In the second paragraph of the introduction, the authors mentioned LSBq and NEQR but didn’t define what is an abbreviation of LSBq and NEQR?

3.     Figure is cut n paste image. It should be re-develop using some tool.

4.     In section II, the authors mentioned QPUs as quantum processors and next as graphics processing units. Pl. differentiate between these two.

5.     The equations shown in the paper are not indexed. P. index all the equations.

6.     Figure 4 is not visible properly. It should be redesigned.

7.     In table 1, the Methodology and Outcomes are not properly formatted.

8.     Imperial data is missing in this paper.

9.     Future work is not mentioned in the conclusion.

10.  Reference no. 1,3,20,23,30,32 are too old.

Author Response

We’re very thankful for taking the time to review our paper and provide us your kind feedback. We’re more than glad to hear your thoughtful suggestions to improve our manuscript. We hope that the updated version of our paper lives up to your standards. Our rebuttal is elaborated below:

  1. We’re grateful for your observation and we highly agree with this suggestion. We have removed “Recent” from the manuscript’s title page.
  2. Thank you for your great observation. Upon your feedback, we have provided further elaboration for both LSBq and NEQR and their definition in our paper.
  3. Thank you for your thoughtful insight. We have provided a higher resolution version from all the figures in our manuscript. We have also re-developed some of the lower resolution figures in our paper.
  4. Thank you for pointing out this remark. We have cleared the misconception between QPUs and GPUs. To elaborate, QPUs stands for Quantum Processing Unit, while GPU is an acronym for Graphics Processing Unit.
  5. We appreciate your observation. We have numbered all the formulas in the newer version of the manuscript.
  6. We’re grateful for this observation, we have increased the resolution for all the figures in our manuscript.
  7. Thank you for your thoughtful feedback, we have updated the formatting for both methodology and outcomes section of table 1.
  8. We deeply appreciate your kind suggestion. We'd like to clarify that since our manuscript is considered as a survey paper, it is difficult to include empirical data into survey articles.
  9. Thank you for your suggestion. We hope you understand that the old references me paper are all necessary in our manuscript since they are considered as a vital part of the referenced scheme’s brief background description. And that it would be difficult to leave them out from their sections. However, we did remove old references from the introduction section and excluded any unnecessary old references.

Reviewer 3 Report

This paper provides an overview of the latest advances in the field of quantum steganography and image steganography schemes. However, the following comments are improved and should be considered in this round.

In the general, the new section’s consistent style is missing in entire the paper. Thus, the authors are recommended to rewrite the paper by drawing the LATEX tool (overleaf for simplicity).

Authors are recommended to redraw Figure 1 in order to increase quality.

In line 70, authors are recommended to replace the section with BACKGROUND and provide in with classical steganography, the system model (adding figure), participating components, potential attacks, etc.

In line 152, the authors claim “In this section, we discuss eleven quantum steganography techniques”, while figure 1 only included 10 techniques.

In line 124, the description of the techniques should be included in detail. Please move figure 1 to this section after revising.

Table 1 should be included the limitations as a new column.

Discussion  of this paper is missing

Comparative this survey with other surveys is missing.

Poor conclusion and abstract.

Authors are recommended to follow the presentation of this paper according to the following paper entitled Survey of Authentication and Privacy Schemes in Vehicular Ad Hoc Networks

Author Response

We’re very grateful for taking the time to review our paper and providing us your thoughtful insights for improving our work. Hopefully the updated version lives up to the required standards. Our rebuttal is elaborated below:

  1. We’re thankful for your thoughtful suggestion. And upon your feedback, we have applied the LATEX tool to improve our work and the structure of our paper.
  2. Thank you for the observation. We have redrawn figure 1 to make it more visually clear upon your suggestion.
  3. We’re very grateful for the great observation. Based on your suggestion, we have changed the title for section 2 into “BACKGROUND” and have incorporated a short segment explaining the security risks and potential attacks that steganography schemes might face along with the history of steganography, and the most commonly used quantum image representations for applying steganography.
  4. Thank you for your observation. We have fixed the error and now it states “ten” rather than “eleven” in the manuscript.
  5. We’re grateful for your suggestion. The reason why we added figure 1 in the first section was to make sure the reader can have a clearer picture of the paper’s overall structure after reading the introduction, making it easier to follow up with the following sections.
  6. We appreciate your thoughtful suggestion. We have added a new section into table 1 named “limitations” upon feedback, where we added the limitations faced by each mentioned scheme in the table.
  7. We’re grateful for this suggestion. However, we found it difficult to incorporate a discussion section into our paper since it’s aimed to be a straightforward survey article which aims to gather and discuss the latest advances presented in the quantum image steganography field.
  8. Thank you for your thoughtful observation, and we believe that creating a comparative study between our survey paper and other survey papers will strengthen our manuscript. But due to the lack of survey papers discussing quantum image steganography schemes, we couldn’t apply this suggestion in our paper. Hopefully our paper could encourage future researchers to invest more into quantum steganography.
  9. We’re thankful for your observation. We have updated the abstract and conclusion section of our paper upon your feedback.
  10. We appreciate your kind suggestion to improve our paper, and we’re grateful for your recommendation. We have read and were inspired by the article titled: “entitled Survey of Authentication and Privacy Schemes in Vehicular Ad Hoc Networks”. We have also got inspired by some of its elements. Therefore, we decided to add it into the list of references for our paper.

Round 2

Reviewer 3 Report

However, the revision was not sufficient for the authors. The paper still suffers from structuring, In LATEX, this error should not appear. Add a little space for new any paragraph such as in lines 18,48,54. In figure 1, the caption is not attached directly. Thus, this proves that the authors don’t use the LATEX tool.

Please add a new section called discussion and show future challenges, 

Author Response

Thank you for taking the time to review our manuscript resubmission, and we are thankful for your kind insight into improving our work. We hope that this version of the manuscript satisfies your expectations of our work.

Point 1:

However, the revision was not sufficient for the authors. The paper still suffers from structuring, In LATEX, this error should not appear. Add a little space for new any paragraph such as in lines 18,48,54. In figure 1, the caption is not attached directly. Thus, this proves that the authors don’t use the LATEX tool.

Rebuttal:

We are very grateful for your observation, and we agree that the structure of the manuscript needed improvement through LATEX. Therefore, we have provided a new version of the manuscript using LATEX. We have also applied MDPI’s LATEX template into our work to eliminate any design errors and match our manuscript to MDPI’s requirements. We thank you again for your suggestion and we hope that our new version of the manuscript is satisfactory.

Point 2:

Please add a new section called discussion and show future challenges,

Rebuttal:

Thank you for your kind suggestion. Following by your request, we have incorporated a new section to the document titled “Discussion” section in the updated version of the manuscript, where we have discussed the points mentioned in the previous sections along with future directions.